# Fabry–Perot Effect Suppression in Gas Cells Used in THz Absorption Spectrometers. Experimental Verification

**DOI:** 10.3390/s24227380

**Published:** 2024-11-19

**Authors:** George K. Raspopin, Alexey V. Borisov, Arnaud Cuisset, Francis Hindle, Semyon V. Yakovlev, Yury V. Kistenev

**Affiliations:** 1Laboratory of the Molecular Imaging and Machine Learning, Tomsk State University, 634050 Tomsk, Russia; raspopingk@mail.tsu.ru (G.K.R.); borisov@phys.tsu.ru (A.V.B.); 2Laboratoire de Physico Chimie de l’Atmosphère, Université du Littoral Côte d’Opale, 59140 Dunkirk, France; cuisset@univ-littoral.fr (A.C.); francis.hindle@univ-littoral.fr (F.H.); 3V.E. Zuev Institute of Atmospheric Optics Russian Academy of Sciences, Siberian Branch, 634055 Tomsk, Russia; ysv@iao.ru

**Keywords:** amplitude modulation spectroscopy, Fabry–Perot effects, frequency modulation spectroscopy, terahertz gas absorption spectroscopy

## Abstract

A standard measuring gas cell used in absorption spectrometers is a cylinder enclosed by two transparent windows. The Fabry–Perot effects caused by multiple reflections of terahertz waves between these windows produce significant variations in the transmitted radiation intensity. Therefore, the Fabry–Perot effects should be taken into account to correctly measure absorption spectra in Bouguer law-based absorption spectroscopy. One approach to reducing the Fabry–Perot effects is based on inserting an additional external movable window with the standard measuring gas cell. This was proposed and numerically analyzed in our previous work. This paper is aimed at the experimental validation of this method when using amplitude modulation (AM) spectroscopy. Also, a comparison of the efficiency of reducing the Fabry–Perot effects using this method is experimentally compared to frequency modulation spectroscopy. The latter was shown to effectively reduce the Fabry–Perot effects compared to AM spectroscopy with the standard measuring gas cell, and the use of the external movable window was shown to further improve the elimination of Fabry–Perot effects.

## 1. Introduction

The standard measuring gas cell used in gas Bouguer law-based absorption spectroscopy is a cylinder oriented along the propagation direction of the sensing optical radiation beam constrained by two transparent windows. The studied gas sample absorption coefficient α(ω) is expressed as follows:(1)$$\alpha \left(\omega_{c}\right)=-\frac{\ln\left(T\left(\omega_{c}\right)\right)}{L},$$
where $T\left(\omega_{c}\right)=I/I_{0}$—the transmittance of the measuring gas cell filled by studied gas sample; $I_{0},I$—the intensities of the sensing optical wave in the input and output of the measuring gas cell respectively; L—this cell length; and $\omega_{c}$—frequency of the optical wave. The absorption spectrum $I_{D}\left(\omega \right)$ recorded by the detector not only depends on the gas sample absorption but also on the transfer function of the measuring gas cell $T_{FPI}\left(\omega \right)$, as follows:(2)$$I_{D}\left(\omega \right)=I\left(\omega \right)*T_{FPI}\left(\omega \right),$$
where ∗—convolution operator.

The Fabry–Perot (FP) effects caused by multiple reflections of the optical wave between the measuring gas cell windows and in the windows themselves produce a significant impact on the transfer function $T_{FPI}\left(\omega \right)$ [1,2,3]. As a result, the measured absorption spectrum of the gas sample studied may be distorted [4,5,6,7], which significantly reduces the limit of detection (LOD) of the spectrometer [8]. An example of the influence of these effects on the measured gas sample absorption spectrum shape for a standard two-window measuring gas cell is shown in Figure 1. Under specific thermodynamic conditions of the gas sample and cell parameters, the FP effects may not only cause substantial distortion of the absorption line shape (Figure 1a) but also entirely obscure it (Figure 1b).

Possible approaches for FP effect suppression include the use of multilayer antireflection (AR) coatings based on nanostructures [9], polymers [10], meta materials [11,12], and sub-wavelength structures [13,14]. However, the complexity of design and manufacturing limits their application to laboratory prototypes. Time averaging of the amplitude of periodic interference fringes via phase-tuning using an oscillating output window [15] or an optically transparent plate [16,17] placed on a piezoelectric actuator is limited by the range of possible spatial displacement of this actuator (not more than a few hundred micrometers). This method is effective when the latter exceeds several wavelengths of the sensing optical wave. Therefore, implementation of this method in the terahertz (THz) spectral range requires a spatial displacement value of about several millimeters. Tilting of the cell’s input and output windows with the Brewster angle [18] can effectively suppress the interference fringes resulting from reflections solely between the measuring cell windows, but it causes a beam displacement from the initial optical axis. Depending on parameters of the windows, this displacement can achieve several mm. Moreover, this method is not effective for circularly polarized or non-polarized waves.

The frequency modulation (FM) technique (single- or dual-tone) [1,3] has proven its efficiency in reducing the FP effects on absorption line shape. However, the application of FM is suitable mainly in the case when the absorption lines have a gradient many times higher than the slowly varying $T_{FPI}\left(\omega \right)$. Moreover, FM leads to a loss of information about the absorption profile’s original shape, complicating quantification of the target species [19].

Thus, despite significant progress in the field of THz absorption spectroscopy, the problem of suppressing unwanted Fabry–Perot interference effects is still relevant.

This paper is aimed at the experimental verification of the proposed method of reducing the FP effects [20] in a variant of AM THz spectroscopy. Also, the results of an experimental comparison of this method and FM spectroscopy regarding the efficiency of FP effect suppression are also presented.

## 2. Materials and Methods

The $T_{FPI}\left(\omega \right)$ of the standard measuring gas cell with plane-parallel non-absorbing windows can be presented in the following form [21,22]:(3)$$T_{FPI}={\left[1+F\cdot {sin}^{2}\left(\frac{2\omega nd\;\cos\left(\theta \right)}{c}\right)\right]}^{-1},$$
where d—the window thickness, n—window refraction index, θ—the angle of incidence of the optical wave on the window surface, and F—the “finesse” of a measuring gas cell, showing the narrowness of the interference band (in terms of full width at half maximum, FWHM) in relation to its free spectral range (FSR) (Figure 2) [23]:(4)$$F=\frac{FSR}{FWHM}=\frac{\pi }{\arccos\left(\frac{2r_{1}r_{2}}{1\;+\;r_{1}^{2}r_{2}^{2}}\right)}\;,$$
(5)$$FSR=\frac{c}{2nd},$$
(6)$$FWHM=\frac{FSR}{\pi }\arccos\left(\frac{2r_{1}r_{2}}{1\;+\;r_{1}^{2}r_{2}^{2}}\right)\;,$$
where $r_{1},\;r_{2}$—the Fresnel reflection coefficients on the “air–window” and “window–air” boundaries, respectively. A significant distortion of the spectrum by the FP effects is observed when the FSR or FWHM is comparable with the absorption line width.

Therefore, to measure the correct absorption spectrum shape in Bouguer law-based absorption spectroscopy, an optimization of the measuring gas cell’s geometric parameters is necessary in the spectrometer’s operational frequency range.

### 2.1. Fabry–Perot Suppression Methods

Frequency modulation (FM) spectroscopy is often preferred to amplitude modulation (AM) spectroscopy in order to reduce baseline oscillations due to the FP effects [19,24,25].

In FM spectroscopy, a carrier frequency of the optical wave is modulated harmonically: ${\omega \left(t\right)=\omega }_{c}+\delta \omega \cos\left(\omega_{m}t\right)$ with $\omega_{m}\ll \omega_{c}$, where $\omega_{m}$, δω—the modulation frequency and depth respectively. The efficiency of this technique is limited to the condition FWHM>2∆ω, where ∆ω is the absorption line half-width [1]. Assuming a small absorbance ($\alpha \left(\omega \right)L$ ≤ 0.05), the intensity of the FM optical wave transmitted through the measuring gas cell is proportional to the *n*-th harmonic of the Fourier series expansion [26,27]:(7)$$I\left(\omega_{c}+\delta \omega \cdot \cos\left(\omega_{m}t\right)\right)\propto I_{0}\left\{\sum_{n=0}^{\infty }H_{n}\left(\omega_{c}\right)\cos\left(n\omega_{m}t\right)\right\}L$$
where $H_{n}\left(\omega_{c}\right)$—the amplitude of the *n*-th harmonic. To improve the signal-to-noise ratio (SNR), the second harmonic (n = 2) is usually used [19]. For the absorption line described by the Lorentz function, the corresponding second harmonic signal component has the following form [26,27]:(8)$$H_{2}\left(x,m\right)=\frac{4}{m^{2}}-\frac{2^{\frac{1}{2}}}{m^{2}}\frac{\left(M+1-x^{2}\right){\left[{\left(M^{2}+4x^{2}\right)}^{\frac{1}{2}}+M\right]}^{\frac{1}{2}}+4x{\left[{\left(M^{2}+4x^{2}\right)}^{\frac{1}{2}}-M\right]}^{\frac{1}{2}}}{{\left(M^{2}+4x^{2}\right)}^{\frac{1}{2}}}$$
where $M=1-x^{2}+m^{2}$, $x={(\omega }_{c}-\omega_{0})/∆\omega$, $\omega_{0}$—the central frequency of the absorption line, and m=δω/∆ω—the modulation index. The same function for the Voigt or Gaussian profile of the absorption line is presented in [27,28]. As was shown experimentally [19], incorrect choice of the modulation index m leads to significant distortion of the second harmonic profile: if δω≪∆ω—the distortion is negligible, when δω≫∆ω—the line profile is significantly “blurred.” According to Equation (8), the maximum amplitude of the detected signal is achieved when m = 2.2 (Figure 3). Therefore, in the experiment, the modulation depth should be δω≥2∆ω [27,28].

An alternative approach was numerically studied in our previous work [20]. It consists of inserting an additional external movable optically transparent window in front of the standard measuring gas cell. The window should be moved and aligned along the THz beam propagation axis. The initial position of the window can be arbitrarily selected. For a given frequency ω, the window is translated step by step to cover an interval $L_{in}>\lambda$ (λ—THz wave wavelength). The transmitted intensity is recorded for each mirror position l, and the maximal value ${max}_{l\in L_{in}}(I(\omega ,l))$ is retained for this frequency. This procedure is repeated for all frequencies of interest.

To exclude the measurement of any absorption of the windows or the ambient air, the measurements should be repeated with a reference (non-absorbing) gas sample. Thus, the resulting absorption coefficient $\alpha \left(\omega \right)$ can be expressed in the following form:(9)$$\alpha \left(\omega \right)=-ln\left(\frac{\;\underset{l\in L_{\mathrm{in}}}{\max}\left(I_{ref}(\omega ,l)\right)}{\underset{l\in L_{\mathrm{in}}}{\max}\left(I_{D}(\omega ,l)\right)}\right)/L$$
where $I_{ref}\left(\omega ,l\right)$—the intensity of the THz wave passing through the measuring gas cell filled with a non-absorbing gas sample.

### 2.2. Experimental Setup

The experimental setup used in this work is shown in Figure 4. An AnaPico RFS40 frequency synthesizer (AnaPico, Opfikon, Switzerland) (output frequency range 100 kHz–40 GHz, spectral resolution 10^−3^ Hz) combined with a Ceyear 82401U frequency multiplier operating in the 500–750 GHz frequency range (near 3 mW output power) and a standard linear polarization gain horn antenna (WR1.5) with ±8° divergence angle (in the E and H planes) and gain factor ≥ 25 dB were used. A zero-bias Schottky diode detector (ZBD) (Virginia Diodes Inc., Charlottesville, VA, USA) with a diagonal horn (WR1.5, gain factor ≥ 25 dB) combined with an SR510 low–noise single phase lock-in amplifier (Stanford Research System, Sunnyvale, CA, USA) with a typical noise ≤ 7 nV/Hz^1/2^ was used. This detector has the sensitivity of 1000 V/W with the noise equivalent power ≤ 20 pW/Hz^1/2^. The lock-in amplifier output signal was captured by the two-channel 12–bit NI PXI–5124 digital oscilloscope operating with a real-time sample rate of up to 200 × 10^6^ samples per second and 0.65% DC accuracy.

The measurement cell was composed of cylindrical vacuum-tight housing with a length of L = 1 m and an internal diameter of 23 mm. A dedicated gas outlet was connected to a vacuum system to evacuate the measurement chamber. A gas inlet allowed the target gas to be introduced. Both the cell windows and the external movable window were made from polytetrafluorethylene (PTFE or Teflon) (n = 1.46), which is optically transparent in the sub-THz range. The thickness of the external moving window was 1 mm; the thickness of the measuring gas cell windows was 5 mm. Since the objective of this research was to investigate solely the influence of the external window on the resulting absorption spectrum, no AR coatings were used. The choice of thickness of the external window was based on Equations (5) and (6) to minimize the impact of the FP effects between its reflecting surfaces and have FSR and FWHM much larger than the spectral width of the absorption line (2∆ω). For example, an external window made from PTFE of 1 mm thickness provides FSR = 102.7 GHz and FWHM ≈ 62 GHz, while the gas cell of 1 m length with windows made from PTFE of 5 mm thickness has FSR = 0.15 GHz, FWHM ≈ 0.078 GHz, and 2∆ω = 6.75 GHz for the water vapor absorption line with a central frequency of 557.146 GHz at *p* = 1 atm.

The scanning time depends on the parameters of the detection system and the number of points required for recording the absorption line profile and the FP interference patterns accurately. For the used spectrometer, the external window spatial shift of 0.07 mm provided three points in the spectrum subrange corresponding to FWHM ≈ 0.078 GHz and six points in the spectrum subrange corresponding to FSR = 0.15 GHz. When the dwell time at each spectral point was 300 ms and the number of points per absorption spectrum was equal to 500, the total time for its recording was about 10 min.

The external window displacement along the optical axis of the measurement cell was realized by the motorized linear transducer developed and manufactured on a 3D printer (Figure 5). The transducer was controlled by an Arduino Uno board (Arduino LLC., Somerville, MA, USA) connected to a PC via USB interface. The linear transducer provided an external window displacement of up to 200 mm with steps of 62.5 μm.

### 2.3. Experimental Conditions

To study of the efficiency of the proposed method to reduce the FP effects, the water vapor absorption line with a center frequency $\omega_{0}$ = 556.938 GHz was chosen. The humidity of the laboratory air was sufficient to allow this line to be observed easily. The reference signal $I_{ref}\left(\omega ,l\right)$ was measured after cell filling with nitrogen (N_2_) of 99.996% purity. For spectra measured at atmospheric pressure, the range 547.5–565.5 GHz was scanned with a 36 MHz step (500 points per absorption line curve). At a pressure of 0.04 atm, the frequencies 555.5–558.5 GHz were recorded with a 10 MHz step (300 points per absorption line curve). This corresponded to approximately 8 points per interference peak. This was enough to correctly measure FSR between the neighboring interference fringes and their FWHM. Each spectral point was time-averaged (the dwell time at every spectral point was 300 ms), and the lock-in amplifier integration time was 100 ms. Additionally, the signal was filtered by the bandpass filter (30–100 kHz) of a SR560 low noise pre-amplifier (Stanford Research System, Sunnyvale, CA, USA).

## 3. Results

### 3.1. Amplitude Modulation Spectroscopy

A THz wave amplitude modulated by meander with $\omega_{m}$ = 80 kHz modulation frequency and an amplitude of 5 V_p-p_ was used. The external optically transparent window was shifted with a 0.02 mm step for a total distance $L_{in}$ = 0.6 mm. In total, 30 experimental absorption spectra corresponding to 30 positions of the external moving window were measured.

The laboratory air was measured at room temperature (T = 293 K) and two pressures in a measuring gas cell: *p* = 1 atm and *p* = 0.04 atm.

For laboratory air at *p* = 1 atm, the water vapor concentration was 5000 ppm. This concentration corresponded to a relative humidity level of 21.5%. The theoretical curve for these conditions was calculated using the 2020 HITRAN spectral database [29].

For normal pressure (*p* = 1 atm), in the case of a standard measuring gas cell, the FP effects essentially distorted the absorption line shape, “doubling” the absorption peak (see orange line in Figure 6), which in turn led to difficulties in determining the absorption line central frequency $\omega_{0}$ and its half-width ∆ω. The influence of parasitic interference on the absorption line profile shape and the peak value can be drastically reduced when using the external movable window (see blue line in Figure 6).

Measured and theoretical (from 2020 HITRAN spectral database [29]) spectral parameters of the absorption line, such as $\alpha \left(\omega_{0}\right)$, 2∆ω, and $\omega_{0}$, and the errors of these parameters for both the standard measuring gas cell and the same cell with the additional external movable window are shown in Table 1.

The results of the water vapor absorption line spectrum measurements for *p* = 0.04 atm and T = 293 K are shown in Figure 7.

For this pressure, 2∆ω≈ 0.271 GHz and more noticeable changes were observed in the measured absorption line profile using the standard measuring gas cell because FWHM and FSR were comparable with the absorption line full-width. The FP effects caused a noticeable dip in the central part of the spectral curve that almost excluded the possibility of adequate assessment of the gas sample concentration. The proposed variant of the measuring gas cell provided a much smaller distortion of the absorption line spectral curve. The estimated absorption line parameters and relative errors for *p* = 0.04 atm are shown in Table 2.

It is worth making a remark regarding the value of the relative error of ∆ω when evaluating the standard measuring gas cell. In fact, there was a dip in the central part of the absorption line profile. The value of 2∆ω = 0.18 GHz and the corresponding relative error of 33.6% in Table 2 refer to the left part of the absorption line profile measured using the standard cell. Therefore, such a low relative error should be considered as a misunderstanding in interpreting the experimental data. The standard measuring gas cell with the external window provided a more accurate evaluation of the absorption line parameters.

The improvement in the absorption profile associated with adding the external movable window to the standard measuring gas cell could be evaluated qualitatively using the spectral curves X,Y proximity criterion χ(X,Y) [30]:(10)$$\chi \left(X,Y\right)=\frac{\sum_{j\;}\left|\left|X_{j}\right|-\left|Y_{j}\right|\right|}{\frac{1}{2}\sum_{j}\left|X_{j}\right|+\left|Y_{j}\right|}$$
where $X_{j},Y_{j}$—absorption coefficients of the compared spectral curves at the same frequency. The lower the χ(X, Y) value, the closer curve X is to curve Y. The calculated χ(X, Y) values are shown in Table 3.

The FP effects are an important factor that limit the absorption spectrometer’s sensitivity. The limit of detection (LOD), in terms of the absorption coefficient, can be estimated using the formula [31]:(11)$$\alpha_{min}\left(\omega_{0}\right)=\frac{1}{L}ln\left(\frac{1}{1-P_{n}/P_{0}}\right)$$
where $P_{0}—$the input power of sensing THz wave, $P_{n}$—the additive noise power in a spectrometer operation spectral range, and L—the length of a measuring gas cell. Considering the distortion of the absorption line shape caused by the FP effects as noise, the estimated LOD for both the standard measuring gas cell and the same cell equipped with the external movable window are shown in Table 4.

For the current configuration of the THz spectrometer, implementing the external movable window allowed an increase in the LOD of at least twice in comparison with the standard configuration of the gas measurement cell. It should be noted that the experimental data used for LOD evaluations were acquired at non-optimal conditions.

### 3.2. Frequency Modulation Spectroscopy

To compare the effectiveness of the proposed method in reducing the FP effects in AM spectroscopy and FM spectroscopy with the standard measuring gas cell, measurements of the analyzed water vapor line shape at a concentration ~4320 ppm, *p* = 0.04 atm, T = 293 K, and $\omega_{0}$ = 556.938 GHz were conducted with FM spectroscopy (see Figure 8).

The measurements of the second harmonic signal from the water vapor absorption line were carried out with the following parameters: $\omega_{c}\;$ = 557.1 GHz, δω = 0.691 GHz, and $\omega_{m}$ = 7.2 kHz, with a spectral step of 0.007 GHz. The corresponding second harmonic signal frequency was equal to 14.4 kHz. For the pressure *p* = 0.04 atm, these parameters matched the condition δω≥2∆ω. The absorption spectrum at the second harmonic was obtained as a product of averaging 4 spectra.

The raw data from a lock-in amplifier are shown in Figure 8 (no baseline subtraction). The only exception is the procedure for normalizing the spectra to their maximum for scaling.

The value of χ(X,Y) for the spectral curves presented in Figure 8 equaled to 0.016.

## 4. Conclusions

Parasitic FP effects are a strong negative factor in gas absorption spectroscopy, especially in the THz spectral range. Antireflection coatings of windows used in a measuring cell work well only in a narrow spectral range [32,33]. Orientation of the windows at Brewster’s angle relative to the THz wave propagation direction usually causes an essential astigmatism [34].

The proposed method to reduce the Fabry–Perot effects uses an additional external movable optically transparent window [20]. Variation in the spatial position of this additional external movable window allows measuring the absorbance of a studied gas sample when the influence of the Fabry–Perot effects is minimal. This was clearly confirmed by a comparison of the measurements of the water absorption line shape with a central frequency of 556.938 GHz by AM spectroscopy using the standard measuring cell and the same cell with the additional external movable window (Figure 6 and Figure 7). The external window shift’s optimal step was shown to depend on both parameters of the absorption line half-width, FSR and FWHM.

The choice of AM spectroscopy is reasoned by explicit physical interpretability of the measured spectral data, that is not easy when using FM spectroscopy. For example, the shape of the absorption line profile for FM and second harmonic detection are determined by an approximate expression and indirectly depend on the thermodynamic parameters of the gas sample [25].

A comparison of the efficiency of reducing the Fabry–Perot parasitic effects by FM and AM spectroscopies, conducted in terms of the quantitative spectral curves proximity criterion (Equation (10)), showed that FM effectively reduced the FP effects in comparison with AM spectroscopy implemented using the standard measuring cell but not so effectively as the variant of AM spectroscopy with the standard measuring cell equipped with the external movable window.

## Figures and Tables

**Figure 1 sensors-24-07380-f001:**
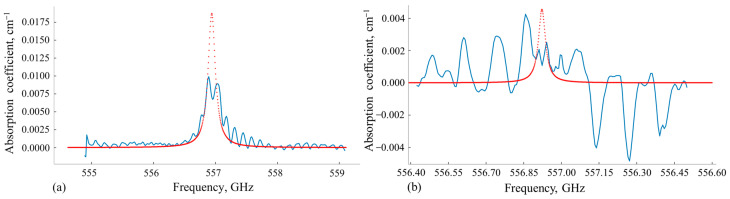
Simulation of the measurements of water vapor absorption spectra near 557 GHz for: (**a**) water vapor concentration ~5000 ppm, *p* = 0.01 atm, T = 293 K; (**b**) water vapor concentration ~1000 ppm, *p* = 0.001 atm, T = 293 K. Red dotted lines—the water vapor absorption coefficient calculated using parameters from the 2020 HITRAN spectral database. Blue lines—water vapor absorption coefficient calculated for standard measuring gas cell with 1 m length, plane-parallel windows of 5 mm thickness, and refractive index n = 2.1.

**Figure 2 sensors-24-07380-f002:**
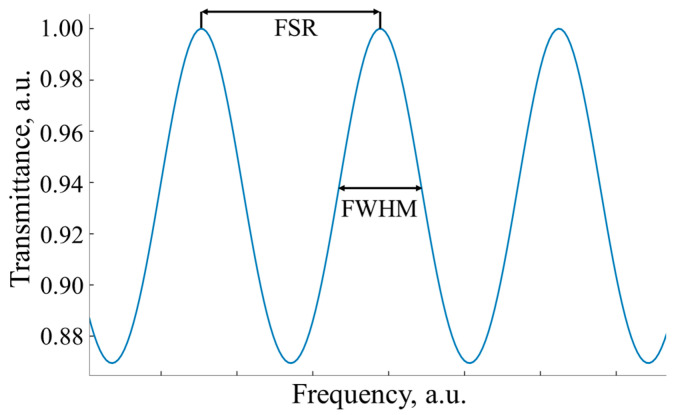
Illustration of FWHM and FSR parameters for a standard measuring gas cell with plane-parallel optically transparent windows.

**Figure 3 sensors-24-07380-f003:**
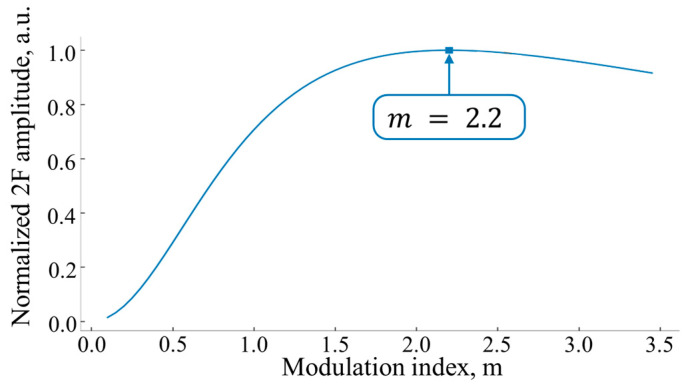
Calculated amplitude of the second harmonic signal (normalized to maximum) as a function of the modulation index m.

**Figure 4 sensors-24-07380-f004:**
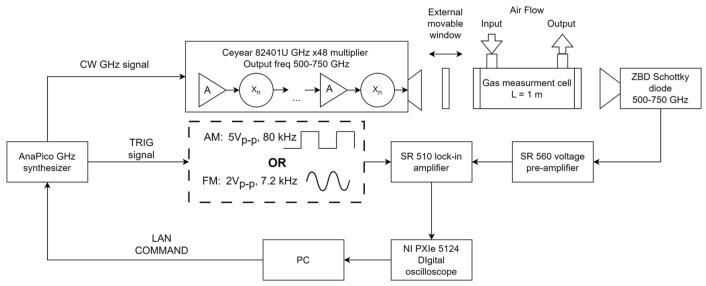
Block scheme of the experimental setup.

**Figure 5 sensors-24-07380-f005:**
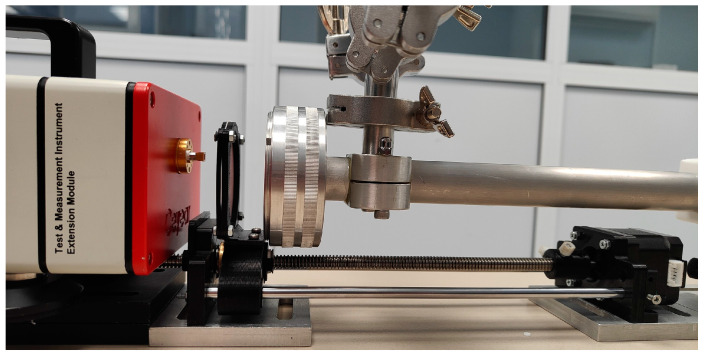
Photo of the measuring gas cell with external moving window placed on the linear transducer.

**Figure 6 sensors-24-07380-f006:**
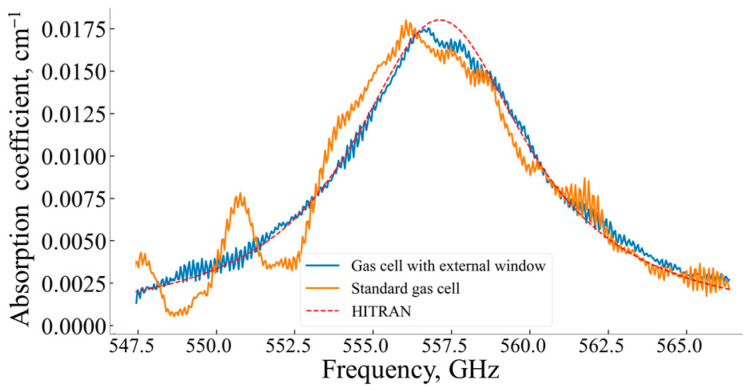
Water vapor line absorption coefficient shape for ~5000 ppm concentration, *p* = 1 atm, T = 293 K. Blue line—water vapor absorption line shape measured in the standard measuring gas cell with additional external movable window, orange line—water vapor absorption line shape measured in the standard gas cell, red dotted line—the water vapor absorption coefficient calculated using the 2020 HITRAN spectral database.

**Figure 7 sensors-24-07380-f007:**
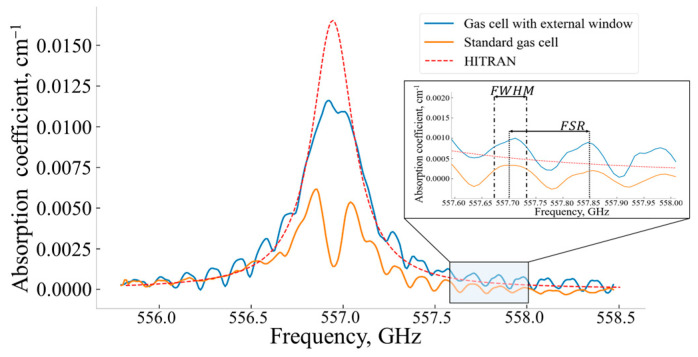
Water vapor line absorption coefficient shape for ~4320 ppm concentration, *p* = 0.04 atm, T = 293 K. Blue line—water vapor absorption line shape measured in the standard measuring gas cell with additional external movable window, orange line—water vapor absorption line shape measured in the standard gas cell, red dotted line—the water vapor absorption coefficient calculated using the 2020 HITRAN spectral database.

**Figure 8 sensors-24-07380-f008:**
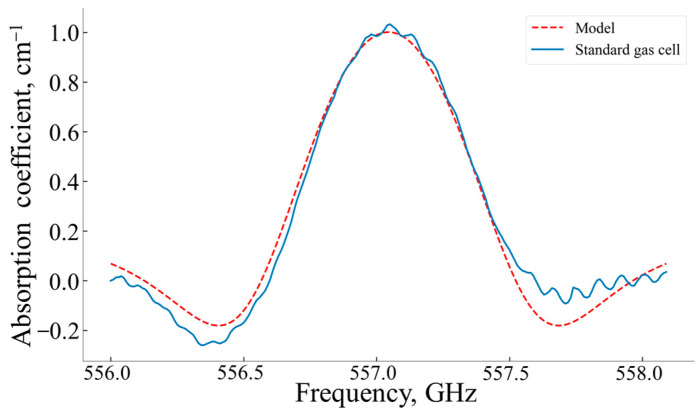
Second harmonic signal for the analyzed water vapor absorption line 556.938 GHz. Blue line—the second harmonic signal measured using the standard measuring gas cell normalized to the maximum value, red line—the same signal calculated for this water vapor absorption line using the 2020 HITRAN spectral database.

**Table 1 sensors-24-07380-t001:** Water vapor absorption line spectral parameters and relative errors of their evaluation for *p* = 1 atm.

	$\alpha \left(\omega_{0}\right)$, cm^−1^	2∆ω, GHz	$\omega_{0}$, GHz
The standard measuring gas cell	0.0185	8.43	556.178
The standard measuring gas cell with additional external movable window	0.0182	6.79	556.764
HITRAN [29]	0.0188	6.75	557.146
	**The relative error, %**		
The standard measuring gas cell	1.6	21.7	0.17
The standard measuring gas cell with additional external movable window	3.2	0.55	0.07

**Table 2 sensors-24-07380-t002:** Water vapor absorption line spectral parameters and relative errors of their evaluation for *p* = 0.04 atm.

	$\alpha \left(\omega_{0}\right)$, cm^−1^	2∆ω, GHz	$\omega_{0}$, GHz
The standard measuring gas cell	0.0061	0.18	556.848
The standard measuring gas cell with additional external movable window	0.0116	0.381	556.915
HITRAN [29]	0.0163	0.271	556.938
	**The relative error, %**		
The standard measuring gas cell	62.57	33.6	0.015
The standard measuring gas cell with additional external movable window	28.83	40.63	0.003

**Table 3 sensors-24-07380-t003:** Values of χ(X,Y) for the standard measuring gas cell and the same cell with additional external movable window.

Gas Sample Pressure	The Standard Measuring Gas Cell	The Standard Measuring Gas Cell with Additional External Movable Window
1 atm	0.063	0.006
0.04 atm	0.711	0.012

**Table 4 sensors-24-07380-t004:** Estimated LOD in terms of the absorption coefficient.

Gas Sample Pressure	$\alpha_{min}\left(\omega_{0}\right)$, cm^−1^ (the Standard Measuring Gas Cell)	$\alpha_{min}\left(\omega_{0}\right),$ cm^−1^ (the Standard Measuring Gas Cell with Additional External Movable Window)
1 atm	1.52 × 10^−3^	0.81 × 10^−3^
0.04 atm	2.89 × 10^−3^	1.38 × 10^−3^

## Data Availability

Data are available upon request to the corresponding author.

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
