# Peer review of "Fabry–Perot Effect Suppression in Gas Cells Used in THz Absorption Spectrometers. Experimental Verification"

_sensors, 2024, doi:10.3390/s24227380_

Round 1
Reviewer 1 Report
Comments and Suggestions for Authors
This paper reduces the Fabry-Perot effect of standard measuring gas cell by inserting an additional external movable window. This method has been numerically studies in their previous work. You experimentally verified this method by using amplitude modulation spectroscopy. However, the article only experimentally verified the previous work and lacked more innovation.
1、 The importance of suppressing the Fabry-Perot effect should be further described in the introduction.
2、 The introduction should include statements of relevant references for suppressing the Fabry-Perot effect.
3、 It may be more appropriate to include a detailed derivation of the Fabry-Perot effect in the methods section, rather than in the introduction.
4、 Can you compare the advantages or disadvantages of this method with other methods experimentally? For example, can it be compared with the gas cell using windows at Brewster angle?
5、 Have you conducted more experiments, such as verifying the results of previous theoretical work?
Author Response
- The importance of suppressing the Fabry-Perot effect should be further described in the introduction.
Answer: Arguments about the importance of suppressing the Fabry-Perot effects have been added in Introduction section.
- The introduction should include statements of relevant references for suppressing the Fabry-Perot effect.
Answer. The statements of relevant references for suppressing the Fabry-Perot effect have been added in Introduction section.
Added relevant references list:
- Decker, J.; Fertein, É.; Bruckhuisen, J.; Houzel, N.; Kulinski, P.; Fang, B.; Zhao, W.; Hindle, F.; Dhont, G.; Bocquet, R.; et al. MULTICHARME: A Modified Chernin-Type Multi-Pass Cell Designed for IR and THz Long-Path Absorption Measurements in the CHARME Atmospheric Simulation Chamber. Atmospheric Measurement Techniques 2022, 15, 1201–1215, https://doi.org/10.5194/amt-15-1201-2022.
- Huang, Y.F.; Chattopadhyay, S.; Jen, Y.J.; Peng, C.Y.; Liu, T.A.; Hsu, Y.K.; Pan, C.L.; Lo, H.C.; Hsu, C.H.; Chang, Y.H.; et al. Improved Broadband and Quasi-Omnidirectional Anti-Reflection Properties with Biomimetic Silicon Nanostructures. Nature Nanotechnology 2007, 2, 770–774, https://doi.org/10.1038/nnano.2007.389.
- Cai, B.; Chen, H.; Xu, G.; Zhao, H.; Sugihara, O. Ultra-Broadband THz Antireflective Coating with Polymer Composites. Polymers 2017, 9, 574, https://doi.org/10.3390/polym9110574.
- Chen, H.-T.; Zhou, J.; O’Hara, J.F.; Chen, F.; Azad, A.K.; Taylor, A.J. Antireflection Coating Using Metamaterials and Identification of Its Mechanism. Rev. Lett. 2010, 105, https://doi.org/10.1103/physrevlett.105.073901.
- Chen, Y.W.; Han, P.Y.; Zhang, X. -c. Tunable Broadband Antireflection Structures for Silicon at Terahertz Frequency. Phys. Lett. 2009, 94, doi:10.1063/1.3075059, https://doi.org/10.1063/1.3075059.
- Simonyan, K.; Gharagulyan, H.; Parsamyan, H.; Khachatryan, A.; Yeranosyan, M. Broadband THz Metasurface Bandpass Filter/Antireflection Coating Based on Metalized Si Cylindrical Rings. Semiconductor Science and Technology 2024, 39, 095012, https://doi.org/10.1088/1361-6641/ad6d86.
- Kim, D.-S.; Kim, D.-J.; Kim, D.-H.; Hwang, S.; Jang, J.-H. Simple Fabrication of an Antireflective Hemispherical Surface Structure Using a Self-Assembly Method for the Terahertz Frequency Range. Lett. 2012, 37, 2742-2744, https://doi.org/10.1364/ol.37.002742.
- Silver, J.A.; Stanton, A.C. Optical Interference Fringe Reduction in Laser Absorption Experiments. Opt. 1988, 27, 1914-1916, https://doi.org/10.1364/ao.27.001914.
- Ohno, S. Fabry-Pérot Interferometer Scanned by Geometric Phase. In Proceedings of the 2018 43rd International Conference on Infrared, Millimeter, and Terahertz Waves (IRMMW-THz), Nagoya, Japan, 09-14 September 2018, https://doi.org/10.1109/IRMMW-THz.2018.8510234.
- Webster, C.R. Brewster-Plate Spoiler: A Novel Method for Reducing the Amplitude of Interference Fringes That Limit Tunable-Laser Absorption Sensitivities. Opt. Soc. Am. B 1985, 2, 1464-1470, https://doi.org/10.1364/josab.2.001464.\
- Wichmann, M.; Scherger, B.; Schumann, S.; Lippert, S.; Scheller, M.; Busch, S.F.; Jansen, C.; Koch, M. Terahertz Brewster Lenses. Optics Express 2011, 19, 25151-25160, https://doi.org/10.1364/oe.19.025151.
- It may be more appropriate to include a detailed derivation of the Fabry-Perot effect in the methods section, rather than in the introduction.
Answer: Done. See Subsection 2.1.
- Can you compare the advantages or disadvantages of this method with other methods experimentally? For example, can it be compared with the gas cell using windows at Brewster angle?
Answer: Done. See Introduction section
- Have you conducted more experiments, such as verifying the results of previous theoretical work?
The previous work was aimed on description of proposed method for the FP effect suppression and its efficiency estimation using numerical simulation of “registration” of the sulfur dioxide absorption band near 660 GHz in the gas mixture consisted of 0.1 % SO2 and 99.9 % N2. The current experimental work studies water vapor absorption line with 557.146 GHz central frequency. In fact, the ambient air was used that allowed us to control real concentration by measuring air humidity. Therefore, water vapor is more convenient substance for experimental study. In any case, our experiments confirmed the usefulness of proposed method.
Thank you for valuable comments.
All corrections are marked by green.
With my best regards Yu.V. Kistenev

Reviewer 2 Report
Comments and Suggestions for Authors
The paper by G.K. Raspopin, et al “The Fabry-Perot effect suppression in gas cells used in THz absorption spectrometers. Experimental verification” describes an experimental verification of a new method proposed by the authors for reducing Fabry-Perot effects by inserting an additional external movable window into a standard gas measuring cell. The article is quite clear and well written, although there are some points that should be clarified before publication in MDPI Sensors.
1. Page 3, lines 85, 86 - “The window can be translation is aligned along the propagation axis of the THz beam.” The sentence is not clear. Do you mean: “The window can be moved and aligned along the THz beam propagation axis.”? Please check and correct.
2. Page 6, line 172 – “and their errors for both”
Do you mean errors compared to the HITRAN spectral database? Please clarify.
3. Page 6, Table 1 – It seems that the relative error, % for the standard measuring gas cell with additional external movable window shouls be 1.6 (not 6.4). Please check and correct.
4. Pages 7, Table 2. This table uses "raw" data. Would it be better to approximate the data with an appropriate line shape before comparing? In this case, the advantages of the setup with an additional external sliding window would be more obvious.
In conclusion, the paper can be published in the journal MDPI Sensors after minor revision.
Author Response
- Page 3, lines 85, 86 - “The window can be translation is aligned along the propagation axis of the THz beam.” The sentence is not clear. Do you mean: “The window can be moved and aligned along the THz beam propagation axis.”? Please check and correct.
Answer. Done. Sorry for the mistake.
- Page 6, line 172 – “and their errors for both”
Do you mean errors compared to the HITRAN spectral database? Please clarify.
Answer. This text was replaced with “these parameters errors for both the standard measuring gas cell and the same cell with the additional external movable window.”
- Page 6, Table 1 – It seems that the relative error, % for the standard measuring gas cell with additional external movable window should be 1.6 (not 6.4). Please check and correct.
Answer: Done. It was corrected. Sorry for the mistake.
- Pages 7, Table 2. This table uses "raw" data. Would it be better to approximate the data with an appropriate line shape before comparing? In this case, the advantages of the setup with an additional external sliding window would be more obvious.
Answer. The idea of the work is to demonstrate abilities of the proposed method of the FP effects reduction, not to find the best way to do it. In this case, it is better to consider the proposed method without a combination with computer methods of raw data processing. This combination will be analyzed in the future works.
In conclusion, the paper can be published in the journal MDPI Sensors after minor revision.
Thank you for valuable comments.
All corrections are marked by green.
With my best regards Yu.V. Kistenev
Reviewer 3 Report
Comments and Suggestions for Authors
The authors proposed an approach to reduce Fabry-Perot effects by inserting an additional external movable window into the standard measuring gas cell of a THz absorption spectrometer. The work aims to experimentally validate this method using amplitude modulation (AM) spectroscopy, and the authors claim that AM spectroscopy is more appropriate than frequency modulation (FM) methods.
However, I have several queries and criticisms regarding the manuscript, which the authors should address to make this paper suitable for publication in this journal.
1. Regarding Figures 5 and 6
The spectrum of the gas cell with the external window (blue line) shows double peaks at 556 GHz. What caused this phenomenon?
2. Regarding page 7, line 15
The authors state, "The values of 2Δ𝜔 = 0.18 GHz and the corresponding relative error of 33.6% in Table 2 refer to the left part of the absorption line profile measured using the standard cell. Therefore, such a low relative error should be considered as a misunderstanding in interpreting the experimental data."
Why is the "left part" referred to in this calculation? Additionally, when comparing Figures 5 and 6, the left part of the standard gas cell spectrum (orange line) appears to be clearly different, even just with a change in pressure. What changed between these two measurements?
3. Regarding equation (10)
Is the following expression, correct?
Please confirm.
4. Applicability of AM and FM methods
Is it possible to apply both AM and FM methods to obtain a more appropriate spectrum?
5. Regarding the external polytetrafluoroethylene (PTFE) window
Did the authors perform any surface treatment on this window? Additionally, why were a thickness of 1 mm chosen? How long did it take to scan for the optimal position?
Thank you for giving me the opportunity to review your research manuscript. I look forward to seeing the revised version.
Author Response
- Regarding Figures 5 and 6
The spectrum of the gas cell with the external window (blue line) shows double peaks at 556 GHz. What caused this phenomenon?
Answer. As it was mentioned in the text, “the FP effects cause noticeable dip in the central part of the spectral curve…”
- Regarding page 7, line 15
The authors state, "The values of 2Δ? = 0.18 GHz and the corresponding relative error of 33.6% in Table 2 refer to the left part of the absorption line profile measured using the standard cell. Therefore, such a low relative error should be considered as a misunderstanding in interpreting the experimental data."
Why is the "left part" referred to in this calculation? Additionally, when comparing Figures 5 and 6, the left part of the standard gas cell spectrum (orange line) appears to be clearly different, even just with a change in pressure. What changed between these two measurements?
Answer:
When the full width at half maximum (FWHM) and amplitude of FP interference peaks are comparable to those of the absorption line, interference effects significantly distort the water vapor absorption line, resulting in noticeable «dip» or «split». When analyzing experimental spectra without prior knowledge of the gas sample’s composition, these "split" peaks may be misinterpreted as absorption lines of individual molecular components.
In fact, when an experimentator will be analyze a such spectral shape, there are no arguments to consider “left part” or “right part” as an analyzed absorption line.
What changed between these two measurements?
It should be taken into account that results in Figs. 5, 6 have different spectral scales. Direct comparison of the experimental absorption spectra for pressure p = 1 atm and p = 0.04 atm is shown in Fig, below. We do not see dramatic differences in the behavior of the curves.

Figure 1.
Blue line– water vapor absorption line shape measured in the standard measuring gas cell with additional external movable window for p = 1 atm, orange line– water vapor absorption line shape measured in the standard gas cell for p = 1 atm.
Green line– water vapor absorption line shape measured in the standard measuring gas cell with additional external movable window for p = 0.04 atm, red line – water vapor absorption line shape measured in the standard gas cell for p = 0.04 atm
- Regarding equation (10)
Is the following expression, correct?
Please confirm.
Answer. This expression has been corrected. Sorry for the mistake.
- Applicability of AM and FM methods
Is it possible to apply both AM and FM methods to obtain a more appropriate spectrum?
Answer:
Yes, it is possible to use both AM and FM and it could be very interesting to do it when we want reduce the baseline oscillation (FM advantage) and to keep direct quantitative information (AM advantage). This strategy was chosen in the LPCA group in the Ref: https://amt.copernicus.org/articles/15/1201/2022/amt-15-1201-2022.pdf
- Regarding the external polytetrafluoroethylene (PTFE) window
Did the authors perform any surface treatment on this window? Additionally, why were a thickness of 1 mm chosen? How long did it take to scan for the optimal position?
Answer: Done
The thickness of the external moving window was 1 mm; the thickness of the measuring gas cell windows was 5 mm. Since the objective of this research was to investigate solely the influence of the external window on the resulting absorption spectrum, no AR-coatings were used. The choice of thickness of the external window was based on Eqs. (5), (6) to minimize the impact of the FP effects between its reflecting surfaces and have and much larger than spectral width of the absorption line (. For example, external window maden from PTFE of 1 mm thickness provides = 102.7 GHz and ≈ 62 GHz, while the gas cell of 1 m length with windows maden from PTFE of 5 mm thickness has a = 0.15 GHz, ≈ 0.078 GHz and = 6.75 GHz for water vapor absorption line with 557.146 GHz central frequency at p = 1 atm.
The scanning time depends on the parameters of the detection system and the number of points required for recording absorption line profile and the FP interference patterns accurately. For used spectrometer, the external window spatial shift of 0.07 mm provides three points in the spectrum subrange corresponding to ≈ 0.078 GHz and six points in the spectrum subrange corresponding to = 0.15 GHz. When the dwell time at each spectral point was 300 ms and the number of points per absorption spectrum was equal to 500, the total time for its recording was about 10 minutes.
Thank you for giving me the opportunity to review your research manuscript. I look forward to seeing the revised version.
Thank you for warm words and valuable comments.
All corrections are marked by green.
With my best regards Yu.V. Kistenev
Round 2
Reviewer 1 Report
Comments and Suggestions for Authors
The authors have revised the manuscript according to the review comments and the manuscript is now suitable for publication
Author Response
Dear Reviewer,
Thank you for support of our work!
Best regards, Yury Kistenev
Reviewer 3 Report
Comments and Suggestions for Authors
The authors revised their manuscript along with reviewers' comments.
The revised manuscript is appropriate and accepted to publish in this journal after a minor revision below.
1. Regarding equation (10)
Is the following expression, correct?

Author Response
Dear Reviewer,
This equation had been corrected. Correction was the adding the modulus for the difference in the numerator. This correction provides a summation of the deviations at each frequency of the two spectral curves.
Now, the numerator under the sign of the sum is the modulus of the difference in the values of the absorption coefficient of the compared spectral curves at a certain frequency. It means that the numerator is a quantification of the absolute total deviation of one spectral curve from another. When we divide the numerator by the denominator, this total deviation by the denominator will become relative and will not depend on the number of points in the spectrum. This criterion is a convenient quantitative assessment of the differences between the two spectral curves. For example, if they match, the criterion value is zero. For two parallel lines, the criterion will be equal to the normalized displacement of one curve relative to the other.
Best regards, Yury Kistenev